# Dose Finding and Food Effect Studies of a Novel Abiraterone Acetate Formulation for Oral Suspension in Comparison to a Reference Formulation in Healthy Male Subjects

**DOI:** 10.3390/pharmaceutics13122171

**Published:** 2021-12-16

**Authors:** Tamás Jordán, Orsolya Basa-Dénes, Réka Angi, János Orosz, Zsolt Ötvös, Andrea Ujhelyi, Genovéva Filipcsei, László Molnár, Tamás Solymosi, Hristos Glavinas, Dominic Capone, Nicola Whitfield, John McDermott, Litza McKenzie, Lauren Shurety, Elizabeth Manning Duus

**Affiliations:** 1Tavanta Therapeutics Hungary Inc., Madarász Viktor utca 47-49, 1138 Budapest, Hungary; tamas.jordan@tavanta.com (T.J.); orsolya.basa-denes@tavanta.com (O.B.-D.); reka.angi@tavanta.com (R.A.); janos.orosz@tavanta.com (J.O.); zsolt.otvos@tavanta.com (Z.Ö.); andrea.ujhelyi@tavanta.com (A.U.); genoveva.filipcsei@tavanta.com (G.F.); laszlo.molnar@tavanta.com (L.M.); tamas.solymosi@tavanta.com (T.S.); hristos.glavinas@tavanta.com (H.G.); 2Tavanta Therapeutics Inc., 201 King of Prussia Rd., Suite 650, Radnor, PA 19087, USA; dominic.capone@tavanta.com (D.C.); nikki.whitfield@tavanta.com (N.W.); 3Quotient Sciences Ltd., Mere Way, Ruddington, Nottingham NG11 6JS, UK; john.mcdermott@quotientsciences.com (J.M.); litza.mckenzie@quotientsciences.com (L.M.); lauren.shurety@quotientsciences.com (L.S.)

**Keywords:** abiraterone acetate, dose reduction, minimizing food effect

## Abstract

Currently approved formulations of the androgen synthesis inhibitor abiraterone acetate (AA) consist of multiple tablets administered daily in a fasted state. Removing the food effect and switching to a suspension formulation is expected to improve the pharmacokinetic profile and facilitate drug administration for patients with late-stage prostate cancer. Two four-sequence, four-period randomized crossover investigations were undertaken to establish the pharmacokinetic profiles of single doses of commercially available Zytiga^®^, as the reference AA (R-AA), and a novel tablet for oral suspension (TOS). Four single doses of TOS (from 62.5 to 250 mg) were compared in study C01, and two single doses each of TOS (250 mg) and R-AA (1000 mg) were compared under fasted and fed (modified fasted for R-AA) conditions in C02. Plasma concentrations of abiraterone over time were measured, and pharmacokinetic parameters were calculated. Each doubling of the dose of TOS was associated with a greater than 3-fold increase in exposure. A single dose of TOS (250 mg) exhibited similar exposure over 24 h, whether given fasted (625 ng × h/mL) or fed (485 ng × h/mL). A single dose of TOS (250 mg) was associated with higher (fasted, *p* = 0.028) or equivalent exposure (fed) compared to 1000 mg R-AA fasted (532 ng × h/mL). Substantially higher exposures were seen with 1000 mg R-AA under modified fasted conditions compared to TOS, irrespective of prandial status (*p* < 0.001). TOS was generally safe and well tolerated in the study. A 250 mg dose of a novel AA formulation for oral suspension demonstrated bioequivalence to 1000 mg R-AA under fasted conditions. This novel TOS formulation also addresses some of the limitations of current AA treatment, including low bioavailability, high variability in systemic exposure and a large food effect. It may offer an alternative for patients with dysphagia or discomfort with swallowing large pills.

## 1. Introduction

Abiraterone acetate is indicated for patients with metastatic castration resistant prostate cancer (mCRPC) and metastatic high-risk castration-sensitive prostate cancer (CSPC) and included in the WHO’s List of Essential Medicines [1]. It is a prodrug which is converted in vivo to abiraterone. Abiraterone inhibits androgen synthesis by selectively inhibiting 17α-hydroxylase/C17,20-lyase which catalyzes the conversion of pregnenolone and progesterone into testosterone precursors. Inhibition of androgen synthesis results in decreased proliferation of androgen sensitive prostate cancer cell lines and prolongs survival in patients with metastatic prostate cancer [2].

Abiraterone acetate was originally approved in 2011 under the trade name Zytiga^®^. The approved oral dose is 1000 mg administered once daily as 4 × 250 mg tablets, or 2 × 500 mg tablets. Each 250 mg tablet of R-AA weighs 0.72 g and measures 15.9 mm long × 9.5 mm wide. The bioavailability of R-AA is highly sensitive to the patient’s prandial status, having up to 10-fold higher plasma concentrations when taken with food. For this reason, R-AA must be taken on an empty stomach; no food should be consumed for at least two hours before the dose of R-AA is taken and for at least one hour after the dose of R-AA is taken [3,4]. This special prandial state is referred to as a “modified fasting state” by FDA.

In 2018, a second drug product containing abiraterone acetate (Yonsa^®^) was approved by the FDA [5]. Yonsa^®^ contains the active ingredient in “fine particle” form to improve the bioavailability of the compound [6], which allows a reduction in the dose to 500 mg. Based on the prescribing information, Yonsa^®^ tablets can be taken with or without food; however, even this advanced formulation still exhibits a considerable positive food effect [7].

Other oral formulation strategies, such as use of amorphous solid dispersion, particle size reduction and salt formation, have also been applied to improve the bioavailability and reduce the pharmaceutical food effect of the compound [8].

Recent clinical studies indicate that there is a relationship between steady state abiraterone trough concentrations (*C_t_*), primary treatment resistance, prostate-specific antigen (PSA) response and progression free survival (PFS) in mCRPC [9,10]. In general, maintaining abiraterone *C_t_* values above 8.4 ng/mL was observed to lead to better responses in PSA values and prolonged PFS [9]. The highly variable PK profile of currently approved formulations of abiraterone may put patients at risk of underexposure. Therefore, an improved formulation of abiraterone with a more predictable plasma exposure should, in principle, lead to more favorable outcomes for patients.

R-AA consists of a micronized crystalline form. Amorphous, non-crystalline forms were shown to greatly enhance the bioavailability of compounds with unfavorable dissolution characteristics [11]. Previously, a lyophilized powder prototype formulation of amorphous abiraterone acetate was developed with improved solubility and dissolution characteristics [12]. Comparing clinical exposures to historic R-AA data, it was predicted that a single 250 mg dose of this novel powder formulation would achieve plasma exposures equivalent to 1000 mg of R-AA in the fasted state and reduce the substantial positive food effect. To obtain a commercially acceptable final dosage formulation, a tablet for oral suspension (TOS) formulation was developed based on the powder formulation. The formulation for suspension might allow a dose reduction and could eliminate the requirement of taking the drug on an empty stomach [13]. It also offers an alternative for patients with dysphagia (difficulty or discomfort in swallowing) or aversion to swallowing large pills. Such properties are of particular importance, since dysphagia reportedly affects 20–40% of cancer patients and can lead to suboptimal adherence to prescribed medications [14,15,16].

Based on current FDA guidance, the bioequivalence of a new abiraterone acetate formulation with R-AA is established if the main pharmacokinetic parameters (AUC, *C_max_*) of the new formulation fall between the lower and upper limits determined for R-AA under currently approved dosing instructions. Specifically, the lower and upper bioequivalence limits for the 90% confidence intervals (CI) of the ratios are defined as 80% and 125%, respectively; i.e., if the 90% CI is contained entirely above 80%, then the TOS is concluded to provide equivalent or higher exposure than R-AA, and similarly, if the 90% CI is contained entirely below 125%, then the TOS is concluded to provide equivalent or lower exposure than R-AA [17]. In this paper, we report the results of a two-part clinical pharmacokinetic study that was designed to characterize the pharmacokinetic properties of TOS and establish the novel formulation’s equivalency to 1000 mg R-AA.

## 2. Materials and Methods

### 2.1. Chemicals

Micronized abiraterone acetate was purchased from Sterling (Cramlington, UK). Polyvinylpyrrolidone was obtained from Ashland (Covington, KY, USA). Soluplus^®^ was purchased from BASF (Ludwigshafen, Germany). SIF powder was purchased from biorelevant.com, Switzerland. FaSSIF V2 biorelevant medium was prepared according to the manufacturer’s instructions. Lactose monohydrate was purchased from Meggle (Wasserburg am Inn, Germany), sodium deoxycholate was purchased for New Zealand Pharma (Palmerston North, New Zealand), microcrystalline cellulose was purchased from JRS Pharma (Rosenburg, Germany) and crospovidone was purchased from BASF (Florham Park, NJ, USA). All other chemicals were purchased from Sigma (St. Louis, MO, USA).

### 2.2. Preparation of TOS

Amorphous compositions of abiraterone acetate and polyvinylpyrrolidone were produced by hot melt extrusion. Milled extrudate was blended with Soluplus^®^, sodium deoxycholate, microcrystalline cellulose, crospovidone, lactose monohydrate and lubricants and then tableted to produce 23 mm tablets.

### 2.3. Reconstitution Test for TOS

The administrability of the formulation was investigated. Four tablets of TOS were thrown into 250 mL of water. Disintegration of the tablets was aided by gentle agitation. Samples were taken at 1, 5, 10 and 15 min. The samples were filtered using 220 nm PTFE-L disposable syringe filters (Nantong Filter FilterBio Membrane Co., Ltd., Nantong, China). Active content of the filtrate was determined by HPLC.

### 2.4. Fasted State Biorelevant Dissolution Tests

Fasted state dissolution of TOS was measured in three stage dissolution tests. The formulae were dispersed in water at 1.0 mg/mL abiraterone acetate concentration and were held in water for 30 min. This represented the dispersion of a human dose in a glass of water. Simulated gastric fluid (SGF) was added to the aqueous dispersion, pH was set to 1.6. After 30 min the pH was neutralized with sodium maleate buffer and FaSSIF V2 solution (pH = 6.5) was added to the system.

Micronized API was used as reference. In this case the first stage of the dissolution method (dispersing the test item in water) was abandoned, the API was added directly to SGF to simulate per os administration.

Dissolution tests were run in triplicate at 37 ± 1 °C on a Digital Heating Shaking Drybath (Thermo Fischer Scientific, Hillsboro, OR, USA). Samples were taken at the indicated time points and were filtered using 220 nm PTFE-L disposable syringe filters (Nantong Filter FilterBio Membrane Co., Ltd., Nantong, China). Active content of the filtrate was determined by HPLC.

### 2.5. Quantification of the Active Ingredient in Solution

Determination of the active ingredient in solution was performed by HPLC using a Shimadzu Nexera XR HPLC system (Shimadzu Europa GmbH, Duisburg, Germany) with a Kinetex EVO C18 100 Å 100 × 4.6 mm 5 µm (Phenomenex, Torrence, CA, USA) column. A gradient elution program was conducted for chromatographic separation with mobile phase A: 0.1% trifluoroacetic acid (ULC/MS Optigrade, Promochem, Wesel, Germany); and mobile phase B: acetonitrile (Lichrosolv Gradient Grade, Merck, Darmstadt, Germany) as follows: initial A:B = 80:20, 0–1 min A:B = 5:95, 1–3.5 min A:B = 5:95, 3.5–4 min A:B = 80:20, 4–6 min A:B = 80:20. The flow rate was 2 mL/min and the injection volume was 5 µL. The overall run time was 6.0 min. The column temperature was set to 30 °C. Abiraterone acetate was detected with an UV detector at 254 nm.

### 2.6. Clinical Study Design

Study C01 (dose proportionality) and study C02 (comparative food effect) were both single-dose, randomized, open-label, four-period crossover studies conducted at a single center at Quotient Sciences (Nottingham, UK). In study C01, adult male subjects were administered 1–4 tablets of TOS (corresponding to 62.5 mg, 125 mg, 187.5 mg and 250 mg of abiraterone acetate) in the fasted state. In study C02, a different group of healthy male subjects was administered 1000 mg R-AA in the fasted state and in the modified fasting state (meal 2 h before administration), which conditions meet the dosing/food instructions that are within the approved label of R-AA, while 250 mg TOS was administered in the fasted state and after an FDA standard high-fat breakfast which was consumed within 30 min prior to dosing.

The studies were conducted in accordance with the Clinical Protocol, the Declaration of Helsinki and its amendments, the International Conference on Harmonization Good Clinical Practice (ICH GCP) Guidelines, and in accordance with all applicable regulatory requirements (EudraCT: 2019-003417-34, date of Ethical Approval: 12 September 2019). TOS was manufactured at Quotient Sciences (Reading, UK) using hot melt extrusion to produce an amorphous composition of abiraterone acetate which was subsequently formulated as a 62.5 mg unit dose TOS. Prior to dosing, the number of tablets required to achieve the target dose were reconstituted in 240 mL of water. R-AA tablets were obtained commercially and were administered orally as four 250 mg tablets swallowed with 240 mL of water. In both studies, venous blood samples of approximately 4 mL each were collected for the determination of plasma concentrations of abiraterone immediately prior to dosing and at 0.25, 0.5, 0.75, 1, 1.5, 2, 3, 4, 6, 8, 12, 16, 24, 48 and 72 h post-dose. The minimum washout period between each consecutive dose was at least 7 days.

### 2.7. Study Population

Healthy subjects were selected by the investigators based on their medical history, physical examination, electrocardiograms and routine clinical laboratory test results. Twelve (study C01) and twenty (study C02) male subjects aged 21–65 years were enrolled (Table 1). All subjects gave written informed consent and received an inconvenience allowance for their participation.

### 2.8. Bioanalytical Method

Human plasma samples were analyzed for abiraterone using a validated liquid chromatography method with tandem mass spectrometry (LC/MS/MS) on an API5500 mass spectrometer (Applied Biosystem) method at LGC Limited (Fordham, Cambridge, UK). Method validation was based upon “Guideline on Bioanalytical Method Validation”, EMA, CHMP, EWP, July 2011 with reference to the “Guidance for Industry, Bioanalytical Method Validation” recommendations issued by the U.S. Department of Health and Human Services, Food and Drug Administration, Center for Drug Evaluation and Research (CDER), Center for Veterinary Medicine (CVM), May 2001, BP.

Aliquots of 50.0 μl prepared standards, QC samples or samples were added to a 2 mL 96 well plate. 20 μl of 0.1% formic acid in acetonitrile was added containing the internal standard [^2^H_4_]-Abiraterone. Mixing was at 1200 rpm for 30 s before a further dilution with 500 μL of 0.1% formic acid in acetonitrile and mixing at 1200 rpm for 1 min. Samples were centrifuged at 2000× *g* for 10 min at 4 °C, with 100 μL of the resultant supernatant transferred to a 96-well plate containing 100 μL of 0.1% formic acid in acetonitrile and mixed at 1500 rpm for 1 min.

The plate was centrifuged at 2000× *g* for 10 min at 4 °C, with the resulting supernatant analyzed by LC/MS-MS. Sample injection volume was 4 µL, and separations were performed with a 2-min gradient elution with mobile phase A: 0.2% formic acid in acetonitrile; mobile phase B: 0.2% formic acid in water on an Acquity UPLC C18 column with 0.5 mL/min flow rate. Typical retention time for abiraterone was 0.9 min.

The bioanalytical method was found to be linear for abiraterone over the calibration range of 0.2–200 ng/mL. The precision and accuracy of the method was found to be within the target limits of within 20% at the lower limit of quantification and within 15% at all other concentrations. The recovery of abiraterone from human plasma was consistent across the analytical range and acceptable and no significant plasma matrix effects were observed.

### 2.9. Pharmacokinetic Evaluation

The plasma concentration–time profile of abiraterone was analyzed using noncompartmental methods with WinNonlin PK software (v6.3, Certara USA, Inc., Princeton, NJ, USA). Parameters determined included the maximal plasma concentration (*C_max_*), area under the concentration–time curve from time 0 to *t_last_* (AUC_0–t_), area under the concentration–time curve (AUC_0–∞_) and apparent terminal elimination half-life (*t_1/2_*). Area under the curve was determined using the linear trapezoidal method.

In study C01, *C_max_*, AUC_0–t_ and AUC_0–∞_ for abiraterone were analyzed to determine dose proportionality using a power model, with terms for dose fitted as a covariate and a random slope and intercept fitted for each subject.

In study C02, *C_max_*, AUC_0–t_ and AUC_0–∞_ were analyzed to determine relative bioavailability of TOS and R-AA and assess the presence of a food effect. PK parameters underwent a natural logarithmic transformation and were analyzed using mixed effect modeling techniques with terms for treatment, period and sequence as fixed effects and subject as a random effect. Adjusted geometric mean ratios (GMRs) and 90% CIs for the adjusted geometric mean ratios were calculated.

### 2.10. Safety Evaluations

Safety was assessed in all subjects through monitoring of changes in vital signs, clinical laboratory test results, physical examinations, ECGs and adverse event reports.

## 3. Results

### 3.1. In Vitro Characterization of the Formula

The results of the reconstitution test can be seen on Figure 1. After 1 min of reconstitution time, 49% of the AA dose was present in the filtrate. The API contents in the filtrates taken at 5, 10 and 15 min were in the range between 90 and 95% of the dose.

The rapid release of AA from TOS can ensure the reliable administration of the substance in the clinical studies.

The microcrystalline AA reference yielded low solubility in the simulated fasted state intestine. In contrast, the TOS showed high apparent solubility throughout the measurement. For at least 75 min, over 60% of the dose passed the 220 nm filter used for the assays (Figure 2).

Based on the in vitro results of TOS, faster and probably more complete absorption of the dose administered was expected compared to R-AA in the clinical studies.

### 3.2. Subject Disposition and Baseline Characteristics

In the C01 (dose proportionality) study, 12 subjects were enrolled. Two subjects did not take all four doses due to adverse events (see safety summary below). In the C02 (food effect) study, 20 subjects were enrolled, and all subjects completed the study. The demographic and baseline characteristics of the volunteers are presented in Table 1.

### 3.3. Pharmacokinetics

Following the administration of the four doses of TOS in the fasted state (C01 study), mean abiraterone plasma concentrations increased rapidly toward the peak, and declined in a bi- or multiphasic manner. Mean *t_max_* occurred within 1 h, and more than dose-proportional increases of *C_max_* and AUC were observed (Figure 3, Table 2). A doubling of the dose of TOS resulted in a slightly more than 3-fold average increase in exposure as measured by *C_max_*, AUC_last_ and AUC_inf_ (Table 3).

In the comparative food effect study (C02), *C_max_* and AUC_last_ for TOS were reduced by a high-fat breakfast by 58.5% and 22.4%, respectively, contrary to R-AA, which increased by 781.9% and 650.4%, respectively (Table 4). As a result, while *C_max_* for TOS was still in the range observed in the R-AA arms, AUC for TOS in the fed condition was slightly (~10%) below the lower limit of the fasted estimate for R-AA (Figure 4, Table 4, Table 5 and Table 6).

In both prandial states, plasma concentrations for TOS at 24 h were below plasma concentrations measured for R-AA at the same time in the fasted state. Plasma concentrations at 12 h, however, were similar to plasma concentrations observed for R-AA in the fasted state. (Figure 5).

### 3.4. Safety

In study C01, six (50.0%) subjects reported a total of 14 AEs after dosing with TOS, and there were no dose related trends in the incidence of AEs. All AEs, except for two AEs in one subject, were mild and unrelated to the investigational product. No AE was reported by more than one subject at any dose level, except for nasopharyngitis and dry lips, which were reported by two subjects each.

Two subjects were withdrawn due to AEs. The first subject was withdrawn from the study due to worsening of a pre-existing abscess that was considered serious and unrelated, and needed to be drained in hospital. The event was on-going at the time of database lock but was reported by the subject to have resolved without requiring specific treatment approximately 2 months after the last dose. The second subject was withdrawn from the study due to a mild maculo-papular rash that was considered possibly related to the investigational product; this event resolved with use of topical E45 cream for 3 days.

In study C02, twelve (60.0%) subjects reported 22 AEs after dosing with TOS or R-AA. The most frequently reported AEs were associated with musculoskeletal and connective tissue disorders; infections and infestations; and nervous system disorders—5 (25.0%), 4 (20.0%) and 3 (15.0%) subjects reported AEs in these system organ classes, respectively. The two most frequently occurring AEs were back pain (four (20.0%) subjects reported a total of four events) and headache (three (15.0%) subjects reported a total of six events). There was no notable difference in the number of subjects reporting or in the frequency or nature of AEs after dosing with 250 mg TOS or 1000 mg R-AA, or among dosing in the fed, fasted and modified fasted states. All AEs were mild, and the majority were unrelated to the investigational product. Six AEs were considered possibly related to administration of TOS and included headaches: three (15.0%) subjects reported headaches on a total of four occasions; there was one report each of mild somnolence and abdominal discomfort, both of which resolved without treatment. All AEs had resolved by the end of the study.

There were no other clinically significant changes reported in hematological, clinical chemistry or urinalysis laboratory values; vital signs’ measurements; physical examinations; or 12-lead ECGs in either study.

## 4. Discussion

The results of these two phase 1 clinical investigations show that the pharmacokinetic profile of the novel TOS formulation of abiraterone acetate, when given in fasted or fed state, falls between the lower and upper limits defined for R-AA in the fasted and modified fasting state, respectively. Under these conditions, a single dose of 250 mg TOS appears equivalent to a single 1000 mg dose of R-AA under fasted conditions. The TOS formulation of abiraterone acetate also has the potential to remove the current re-strictions on food intake associated as specified in the label for R-AA. Moreover, TOS may provide patients with dysphagia or aversion to swallowing large pills with an alternative dosage form that will improve adherence and ease of administration.

Earlier studies have already shown that prandial status is a critical factor influencing the pharmacokinetics of R-AA [4]. The current study also confirmed that taking R-AA in the true fasted state (e.g., after an overnight fast) results in significantly lower abiraterone plasma concentrations compared to the modified fasted state (2 h after a meal). The considerable variability of PK parameters remains a concern for the administration of R-AA and questions its ability to consistently deliver a pharmacological benefit.

Greater than dose-proportional increases in abiraterone exposure were observed with TOS, suggesting that rapid absorption of TOS may saturate metabolism of TOS in the liver as doses increase. Absorption of TOS in the fasted state was also substantially improved, with 4.7-times higher relative bioavailability when compared to R-AA fasted (on a mg to mg basis). The potential mechanism behind the rapid absorption of the drug from the formulation was discussed by the authors in another article [18]. The substantial food effect seen with R-AA was not observed with TOS—administration of TOS is likely to be far more reliable than that of R-AA, whether given with or without food.

At 24 h post-dose, trough concentrations of TOS were lower than those of R-AA, fasted or modified fasted. This was likely a consequence of the more rapid and complete absorption of TOS, and subsequently, faster clearance. In light of clinical data that link trough abiraterone concentrations to testosterone suppression and clinical response, once a day dosing may not be the right regimen for TOS. Further investigation is necessary to determine the appropriate dosing frequency for TOS.

The choice of subjects is a limiting factor in the investigation of the pharmacokinetic profile of abiraterone acetate. Both studies needed to be performed in healthy adult males, with a single dose administration, as an extended suppression of androgen synthesis would be considered unethical. A complete understanding of the pharmacokinetic and pharmacodynamic profile of TOS, therefore, needs confirmation with repeat dosing and in patients with metastatic prostate cancer. The highly standardized study setup, while enabling a comparison of two pharmaceuticals under similar conditions, will also not reflect the situation unique to patients with prostate cancer and their typical dosing behavior.

In general, single oral doses of TOS up to 250 mg in the fasted and fed states were well tolerated in healthy male subjects and there were no deaths, or serious AEs during the study. The overall incidence of AEs was low at all dose levels and the majority of AEs were mild in severity and unrelated to TOS or R-AA.

## 5. Conclusions

A 250 mg dose of the novel abiraterone acetate formulation for oral suspension demonstrated bioequivalence to 1000 mg R-AA tablets under fasted conditions. This novel TOS formulation also addresses some of the limitations of current abiraterone treatment, including low bioavailability, high variability in systemic exposure due to food intake and administration of multiple tablets. The novel dosage form (water dispersible tablet) allows more convenient drug administration in patients with dysphagia or with problems associated with the pill burden of taking four large R-AA tablets.

## Figures and Tables

**Figure 1 pharmaceutics-13-02171-f001:**
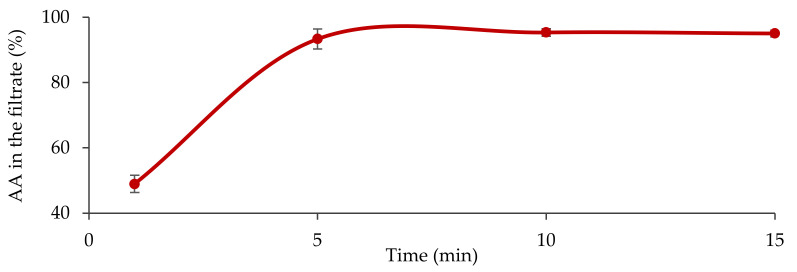
The rate of AA release (mean ± S.D.) during dispersion of 4 tablets of TOS (highest administered dose in the clinical studies) in water.

**Figure 2 pharmaceutics-13-02171-f002:**
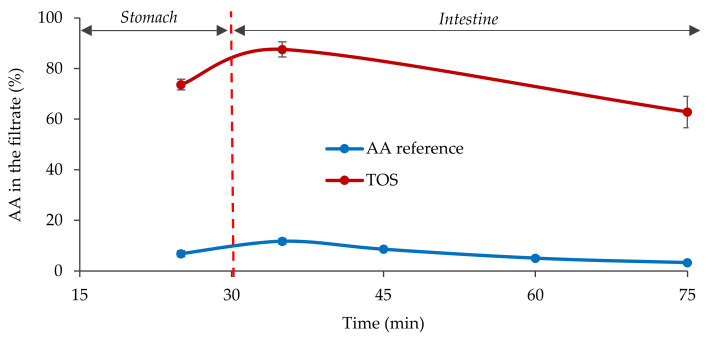
Comparison of fasted state dissolution (mean ± S.D.) of the TOS formulation and the reference AA.

**Figure 3 pharmaceutics-13-02171-f003:**
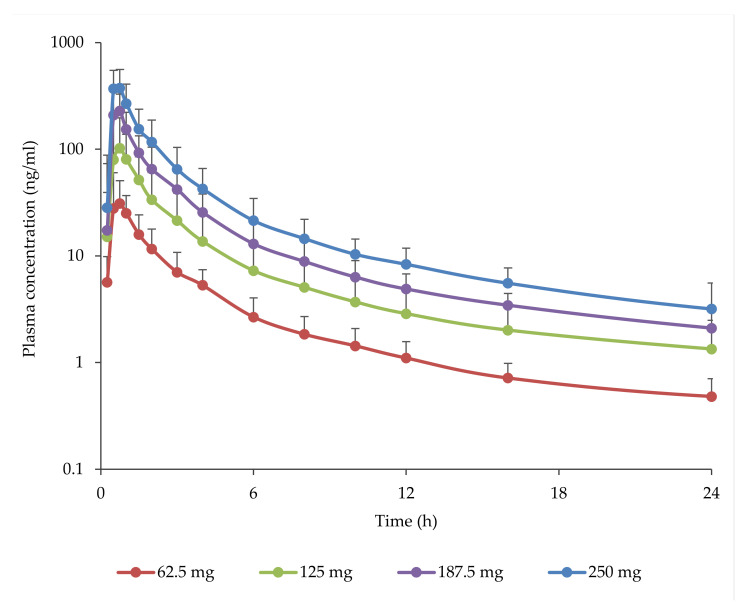
Plasma abiraterone concentrations (geometric mean ± S.D.—only +S.D. are shown) following the oral administration of increasing doses of TOS in the fasted state (study C01, first 24 h). See Table 2 for the number of subjects in each dose level group.

**Figure 4 pharmaceutics-13-02171-f004:**
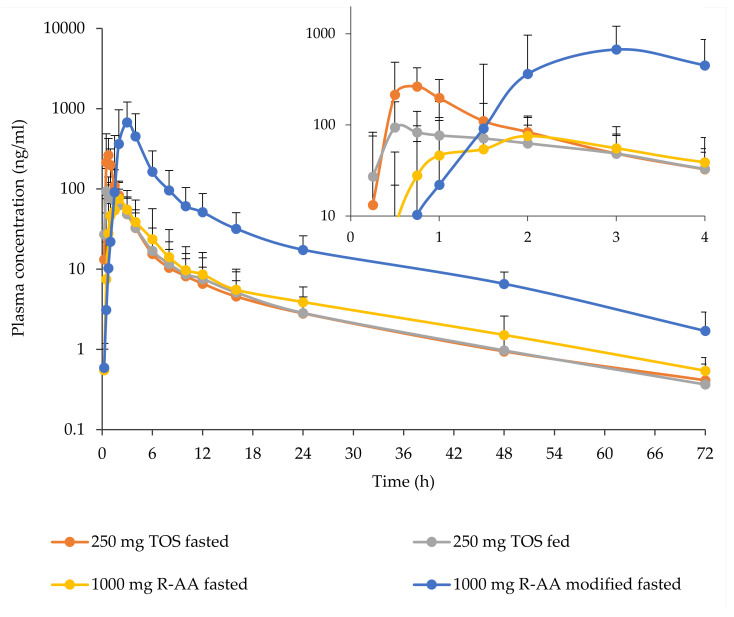
Plasma abiraterone concentrations (geometric mean ± S.D., *n* = 20) following the oral administration of 250 mg TOS or 1000 mg R-AA at different prandial states (study C02). Insert: differences in absorption rate were highlighted by plotting the first 4 h on a separate chart.

**Figure 5 pharmaceutics-13-02171-f005:**
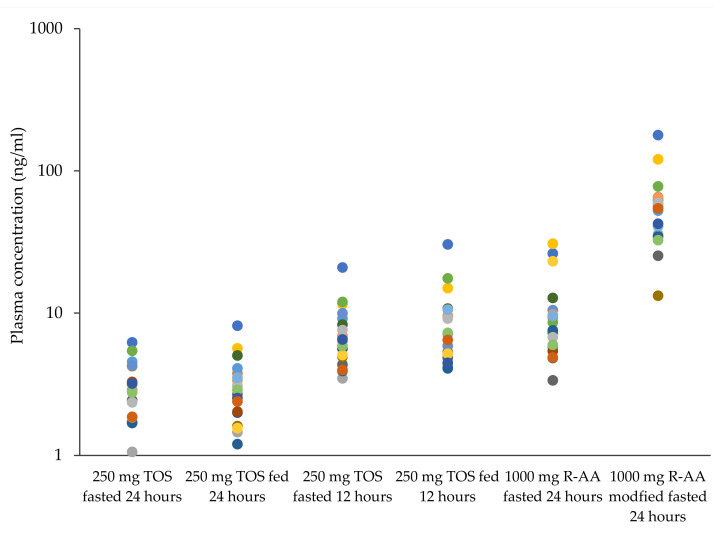
Individual plasma abiraterone concentrations (*n* = 20) following the administration of 250 mg tablets for oral suspension (at 12 h and at 24 h) or 1000 mg R-AA (at 24 h) at different prandial states. The same symbols represent the same subjects.

**Table 1 pharmaceutics-13-02171-t001:** The demographic and baseline characteristics of the subjects in study C01 and C02.

Parameter	Study C01	Study C02
Age (years), mean (range)	46.4 (25–65)	41.0 (21–62)
Race, *n* (%)		
White	12 (100)	18 (90.0)
Black	0 (0)	1 (5.0)
Asian	0 (0)	1(5.0)
Other	0 (0)	0 (0)
Sex, *n* (%)		
Male	12 (100)	20 (100)
Height (cm), mean (range)	176.1 (163–184)	177.5 (166–187)
Weight (kg), mean (range)	79.30 (64.2–93.7)	82.70 (67.8–94.0)
Body Mass Index (kg/m^2^), mean (range)	25.53 (20.2–29.0)	26.27 (22.6–29.2)

**Table 2 pharmaceutics-13-02171-t002:** Pharmacokinetic parameters of plasma abiraterone following the oral administration of increasing doses of tablets for oral suspension in the fasted state (study C01).

Dose Level	62.5 mg	125 mg	187.5 mg	250 mg
No. of Subjects	*n* = 12	*n* = 10	*n* = 11	*n* = 12
*t_max_* ^a^—h (range)	0.50 (0.50 to 1.50)	0.64 (0.50 to 1.50)	0.70 (0.50 to 1.00)	0.75 (0.50 to 0.75)
*C_max_*—ng/mL (CV ^b^%)	43.0 (63.5)	128 (53.9)	265 (49.8)	414 (41.5)
*C_12_*—ng/mL (CV%)	1.10 (37.4)	2.60 (27.8)	4.91 (39.3)	8.31 (40.1)
*C_24_*—ng/mL (CV%)	0.48 (40.1)	1.18 (44.6)	2.10 (47.7)	3.29 (58.1)
*t_last_* ^a^—h (range)	24.03 (23.07 to 48.02)	48.00 (24.30 to 72.18)	70.98 (47.00 to 72.12)	71.01 (48.00 to 72.23)
*C_last_*—ng/mL (CV%)	0.39 (42.6)	0.37 (30.7)	0.41 (41.2)	0.55 (38.8)
AUC_last_—ng × h/mL (CV%)	85.1 (43.2)	242 (37.7)	506 (40.6)	844 (42.7)
AUC_inf_—ng × h/mL (CV%)	91.1 (41.9)	251 (36.9)	516 (39.6)	857 (42.3)
*t_1/2_*—h (CV%)	10.38 (40.8)	14.14 (42.3)	15.04 (29.7)	14.05 (33.4)

Values shown are geometric mean and CV (coefficient of variation), unless otherwise indicated. ^a^ Median (range). ^b^ Coefficient of variation.

**Table 3 pharmaceutics-13-02171-t003:** Assessment of dose linearity.

Dose Level	-	62.5 mg	125 mg	187.5 mg	250 mg
No. of Subjects	-	*n* = 12	*n* = 10	*n* = 11	*n* = 12
*C_max_* (ng/mL):	geometric mean	43	128	265	414
	β (90% CI)	1.64 (1.43, 1.84)			
	2β (90% CI)	3.11 (2.70, 3.59)			
AUC_last_ (ng × h/mL):	geometric mean	85.1	242	506	844
	β (90% CI)	1.65 (1.52, 1.77)			
	2β (90% CI)	3.14 (2.88, 3.42)			
AUC_inf_ (ng × h/mL):	geometric mean	91.1	251	516	857
	β (90% CI)	1.61 (1.49, 1.73)			
	2β (90% CI)	3.05 (2.81, 3.32)			

Results obtained from parametric analysis of log-transformed PK parameters using a power model. The model included terms for dose fitted as a covariate and a random slope and intercept fitted for each subject.

**Table 4 pharmaceutics-13-02171-t004:** Pharmacokinetic parameters of plasma abiraterone following the oral administration of 250 mg tablets for oral suspension or 1000 mg R-AA at different prandial states.

Dose Level	250 mg TOS	250 mg TOS	1000 mg R-AA	1000 mg R-AA
Status	fasted	Fed	fasted	modified fasted
No. of Subjects	*n* = 19 ^b^	*n* = 20	*n* = 20	*n* = 20
t_max_ ^a^ – h (range)	0.50 (0.50 to 1.00)	0.50 (0.50 to 2.00)	1.50 (0.75 to 6.00)	3.00 (2.00 to 4.00)
*C_max_* – ng/mL (CV%)	306 (60.4)	127 (48.0)	105 (58.6)	821 (72.3)
*C_12_* – ng/mL (CV%)	6.80 (44.6) [*n* = 18]	7.55 (57.8)	8.60 (64.3)	51.3 (58.8)
*C_24_* – ng/mL (CV%)	2.80 (46.9)	2.84 (51.1)	3.89 (49.6)	17.4 (52.4)
*t_last_* – h (range)	72.00 (48.00 to 72.08)	72.02 (48.00 to 72.18)	72.00 (48.00 to 72.15)	72.03 (72.00 to 72.47)
*C_last_* – ng/mL (CV%)	0.425 (47.8)	0.372 (41.6)	0.542 (42.9)	1.70 (62.6)
AUC_last_ – ng × h/mL (CV%)	625 (45.7)	485 (44.6)	532 (57.7)	3460 (55.3)
AUC_inf_ – ng × h/mL (CV%)	636 (45.1)	495 (44.0)	547 (56.3)	3510 (55.0)
*t_1/2_*—h (CV%)	15.145 (21.8)	14.624 (22.5)	16.081 (16.5)	13.829 (17.5)

Values shown are geometric mean and CV (coefficient of variation), unless otherwise indicated. ^a^ Median (range). ^b^ One subject had a missing plasma level at 0.75 h, therefore, *C_max_* could not be determined. PK parameters for this subject were not calculated.

**Table 5 pharmaceutics-13-02171-t005:** PK parameters of 250 mg tablets for oral suspension in the fasted and fed states compared to PK parameters of R-AA in the fasted (**A**) and modified fasted states (**B**).

(A)									
			Test		1000 mg R-AA (Fasted)				
Parameter	Test	Prandial State	*n*	Adj Geo Mean ^a^	*n*	Adj Geo Mean ^a^	Ratio ^b^ (%)	90% CI ^c^	*p*-Value ^d^
*C_max_* (ng/mL)	250 mg TOS	Fasted	19	314	20	105	299.83	(238.25, 377.33)	<0.001
AUC_last_ (ng × h/mL)	250 mg TOS	Fasted	19	646	20	532	121.46	(105.18, 140.25)	0.028
AUC_inf_ (ng × h/mL)	250 mg TOS	Fasted	19	657	20	547	120.19	(104.23, 138.59)	0.035
*C_max_* (ng/mL)	250 mg TOS	Fed	20	127	20	105	121.11	(96.61, 151.82)	0.16
AUC_last_ (ng × h/mL)	250 mg TOS	Fed	20	485	20	532	91.24	(79.21, 105.10)	0.28
AUC_inf_ (ng × h/mL)	250 mg TOS	Fed	20	495	20	547	90.46	(78.65, 104.05)	0.24
**(B)**									
			**Test**		**1000 mg R-AA** **(Modified Fasted)**				
**Parameter**	**Test**	**Prandial State**	** *n* **	**Adj Geo Mean ^a^**	** *n* **	**Adj Geo Mean ^a^**	**Ratio ^b^ (%)**	**90% CI ^c^**	** *p* ** **-Value ^d^**
*C_max_* (ng/mL)	250 mg TOS	Fasted	19	314	20	821	38.31	(30.44, 48.21)	<0.001
AUC_last_ (ng × h/mL)	250 mg TOS	Fasted	19	646	20	3460	18.64	(16.14, 21.53)	<0.001
AUC_inf_ (ng × h/mL)	250 mg TOS	Fasted	19	657	20	3510	18.74	(16.25, 21.60)	<0.001
*C_max_* (ng/mL)	250 mg TOS	Fed	20	127	20	821	15.47	(12.34, 19.40)	<0.001
AUC_last_ (ng × h/mL)	250 mg TOS	Fed	20	485	20	3460	14.00	(12.16, 16.13)	<0.001
AUC_inf_ (ng × h/mL)	250 mg TOS	Fed	20	495	20	3510	14.10	(12.26, 16.22)	<0.001

^a^ Adj geo mean = adjusted geometric mean from model, ^b^ Ratio of adjusted geometric means, ^c^ CI = confidence interval for ratio of adjusted geometric means, ^d^ *p*-value from two-sided test with null hypothesis that ratio is equal to 100%.

**Table 6 pharmaceutics-13-02171-t006:** Assessment of food effect with tablets for oral suspension.

			Test		Reference (250 mg TOS) (fasted)				
Parameter	Test	Prandial State	*n*	Adj Geo Mean ^a^	*n*	Adj Geo Mean ^a^	Ratio (%) ^b^	90% CI ^c^	*p*-Value ^d^
*C_max_* (ng/mL)	250 mg TOS	Fed	20	127	19	314	40.39	(32.10, 50.83)	<0.001
AUC_last_ (ng × h/mL)	250 mg TOS	Fed	20	485	19	646	75.12	(65.05, 86.75)	0.002
AUC_inf_ (ng × h/mL)	250 mg TOS	Fed	20	495	19	657	75.26	(65.27, 86.79)	0.002

**^a^** Adj geo mean = adjusted geometric mean from model, **^b^** Ratio of adjusted geometric means, **^c^** CI = confidence interval for ratio of adjusted geometric means, ^d^
*p*-value from two-sided test with null hypothesis that ratio is equal to 100%.

## Data Availability

Data are contained in the article.

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
