# Peer review of "Dose Finding and Food Effect Studies of a Novel Abiraterone Acetate Formulation for Oral Suspension in Comparison to a Reference Formulation in Healthy Male Subjects"

_pharmaceutics, 2021, doi:10.3390/pharmaceutics13122171_

Round 1

Reviewer 1 Report

The reported clinical study is of importance and shows the improved performance (absorption and reduced food effect) of a “novel tablet for oral suspension (TOS)” compared with the currently available product Zytiga.  However, in its current form the manuscript provides limited scientific insight in the field of pharmaceutics - significant improvement are required.

The composition and properties of the “novel tablet for oral suspension (TOS)” needs to be specifically described.  A detailed explanation for its improved performance and mechanism of action needs to be provided.  It would be useful to include solubility data in biorelevant media and discuss with respect to PK data.

The literature review included in the introduction section is considered brief and incomplete.  It should be significantly extended and improved.  The authors should consider recently published reviews on overcoming food effects of abiraterone acetate delivery, e.g.  Schultz et al. “Oral formulation strategies to improve the bioavailability and mitigate the food effect of abiraterone acetate” International journal of pharmaceutics, 577, 119069. https://doi.org/10.1016/j.ijpharm.2020.119069

Reviewer 2 Report

The manuscript by Jordan et al. presents design and result of clinical studies aiming at the evaluate of two formulations of oral abiraterone acetate. Description of he study is adequate and satisfies  methodological requirements of clinical studies. The derived data are analysed and presented according to generally accepted methodology. Consequently, the derived conclusions are well supported by present experimental data. 

For the sake of clarity the abbreviations, in addition to well known parameters as AUC should also include much less recognised parameters presented in Tables 4, 5 and 6 such as AUClast or AUC inf. It would be helpful to define all pharmacokinetic parameters used.

Reviewer 3 Report

The manuscript describes tqp pharmacokinetic studies in healthy subjects, with the purpose to characterize the PK features of a new formulation. The objectives, methods and conclusions are in general sound. A few comment are provided:

  1. Abstract: "pharmacokinetic equivalence" please define or use other wording
  2. Introduction, last paragraph: the text realted to the FDA guidance on bioequivalence is unclear "...is established if the main pharmacokinetic parameters (AUC, Cmax) of the new formulation fall between the lower and upper limits determined for R-AA under currently approved..." Wording needs to be more exact; ie include the accdeptance range for the 90% confidence intervals for the ratio of means between drug formulation, and, include the link to the actual reference for this.
  3. Methods, Clinical study designs: Explain why fed state was not the same for TOS and the original formulation.
  4. Methods, PK evaluation. Please add by which method AUC was calculated.
  5. Results. 3.1: BMI is failry high to reflect healthy subjects - discuss please.
  6. Results. 3.2:"biphacis manner" is not completely clear, would suggest "bi- or multiphacis manner"
  7. Table 4: Frel variables, please add a footnote to explain what is calculated.
  8. Discussion, first paragraph Same comments as point 2 above.
  9. Discussion: Please provide further discussion to why the non-dose proportional exposure is observed in context with the PK characteristics such as oral bioavailability, reason for incomplete oral bioavailability, first pass effect etc. Same comment also with respect to the food effect (formulation effects such as dissolution and solubility).

Round 2

Reviewer 1 Report

The composition and properties of the “novel tablet for oral suspension (TOS)” needs to be specifically described, not just referencing of previous article.  
